# PREDICTING SUBSCRIBER USAGE: ANALYZING MULTI-DIMENSIONAL TIME-SERIES USING CONVOLUTIONAL NEURAL NETWORKS

## ABSTRACT

Companies operating under the subscription model typically invest significant resources attempting to predict customers' future usage. These predictions can be used to fuel growth: Companies can use them to target individual customers – for example to convert non-paying consumers to begin paying for enhanced services – or to identify customers not maximizing their subscription product. This can allow the company to avoid an increase in the churn rate, and to increase the usage of some customers.

In this work, we develop a deep learning model to predict the product usage of a given consumer, based on historical usage. We adapt a Convolutional Neural Network with auxiliary input to time-series data, and demonstrate that this enhanced model effectively predicts future change in usage.

## 1 INTRODUCTION

Historically, companies interested in usage prediction have retained the services of expert analysts. These experts may examine past behavior over multiple features, estimate future usage in the short-term, and translate this knowledge into actionable tasks. Yet for companies which boast thousands of customers and have an increased need for accurate research forecasting, this non-scalable approach quickly becomes infeasible. Hence, the development of an automated model is an obvious necessity (Miller et al., 2021).

Future usage prediction is especially crucial for subscriber-based companies, for which churn rate has especially significant impact on profitability. It is crucial as well for companies valuated based on Annual Recurring Revenue (ARR) which reflects income derived from customers. For these companies, future usage prediction is a prerequisite for predicting the future valuation of the company.

Usage data sets are often represented as multi-dimensional time-series data, and indeed we have at our disposal two large multi-dimensional time-series, which constitute high quality labeled data sets, each produced in a controlled environment. However, successfully predicting multi-dimensional time-series values is itself a non-trivial task (Cerqueira et al., 2019). Indeed, automated multi-dimensional time series prediction is a relatively new field, and although different strategies have been proposed in the literature, currently there seems to be no consensus approach to the problem. Here, machine learning and deep learning algorithms are typically used for time-series data *classification*, which produces a binary decision value (for example, whether the customer usage will grow or not). The problem of *regression*, which produces a continuous value describing customer growth the extent of customer growth, is clearly much more challenging to accurate to accurately predict.

**Our contribution.** For the regression problem of multi-dimensional time-series prediction, we propose to utilize both common machine learning techniques and novel methods. Specifically, we shall leverage one-dimensional convolutional neural network (CNN), in conjunction with auxiliary output, to predict future utilization based on historical usage. In this model, an auxiliary layer produces a prediction for the future in the short-term. This prediction is fed into the next auxiliary layer, which produces a prediction for a future prediction for a slightly later period, until the final auxiliary layer produces a prediction for the desired longer-term period. We demonstrate experimentally that

this enhanced model effectively predicts future usage, and gives superior results when compared to other commonly utilized models in the field.

We present the data sets in Section 3.1, describe our learning model in detail in Section 4, and demonstrate promising empirical results in Section 5. In Section 6, we give concluding remarks and discuss future work.

## 1.1 RELATED WORK

Here we give an overview of some traditional machine learning approaches for time-series prediction (for both classification and regression), as well as more recent CNN-based approaches for this problem.

**Traditional learning approaches.** Tan et al. (2020) aimed to learn the relationship between a time-series and a continuous scalar variable, by identifying the right coefficients to best predict the future variable. Time-series regression models have been applied to various problems such as predicting annual rainfall from observed temperature, or predicting fat content in meat from near-infrared spectrum Goldsmith & Scheipl (2014). That study compared different time series regression models, such as gradient boosting and random forest. It demonstrated that functional regression models usually require an in-depth understanding of the data in order to apply the right basis function to fit the model.

In Pimentel et al. (2015) it was shown that many cutting-edge regression methods fail to distinguish between periods of high and low quality data, and further are not able to generalise well to other data-sets. They attempted to estimate respiratory rates (RR) from a photoplethysmogram (PPG) time-series using a sliding window. Producing an estimation for RR per window consists of four key components: (1) extracting respiratory signals; (2) estimating respiratory rates; (3) fusing the estimates and (4) executing a quality assessments. The authors applied a Gaussian process regression framework to extract RR from the different sources of modulation in the PPG signal, and also tried fitting multiple auto-regressive models to the extracted respiratory signals.

In Zheng et al. (2014), it was shown that $k$-Nearest Neighbor (KNN) combined with Dynamic Time Warping (DTW) achieves state of the art performance for classification. However, although the time complexity of 1-NN with DTW grows linearly with the data size, their performance dependent on the quality of hand-crafted features, and these are not scalable.

In Samsudin et al. (2010), Support Vector Machine (SVM) were compared with Artificial Neural Network ANN on five distinct one-dimensional time-series for the task of regression. They utilized a grid-search technique to identify the optimal hyper-parameters in the data, and used 10-fold cross validation while training. They indicated that SVM was more efficient than ANN on single-dimensional time-series.

**CNN-based approaches.** In Yang et al. (2015) it was shown that classification using CNN on multi-channel human activity recognition time-series outperformed all other algorithms tested. They proposed a systematic feature learning method, adopting CNN to automate feature learning from raw inputs. In deep architecture, the learned features are the high level abstract representations of low level raw time series signals. By leveraging the labelled information through supervised learning, the learned features are presented with more discriminatory power. LeCun et al. (1995) suggested a potentially more interesting scheme (for classification as well), by eliminating the feature extractor, and feeding the network with raw inputs (e.g. normalized images). This relies on back-propagation to turn the first few layers of the neural net into an appropriate feature extractor.

According to Okita & Inoue (2017), the combination of RNN (such as LSTM) and CNN for classification problems such as activity recognition yielded improved accuracy on the range of 27% to 43% when compared to the other popular algorithms. Wang et al. (2019) also considered classification for Human Action Recognition (HAR), and evaluated multiple models, including LSTM, CNN, Random Forest, J48 Decision Tree, and SVM. They found that the superior model was Bidir-LSTM-CNN, a combination of the outputs of a bidirectional LSTM and CNN connected to three fully connected hidden layers.

There has been significant recent interest in analyzing time-series in order to effectively allocate company resources. Snow (2020) undertook an extensive review of popular algorithmic solutions for regression problems, such as Facebook's Prophet. This tool is given a list of dated events, and takes into consideration holidays, weekends and special occasions, before outputting predictions for future usage. Amazon's GluonTS and Microsoft's ForecastTCN attempt to do the same. This problem has also received attention from academia and independent developers such as N-Beats, Auto-Arima, and TBATS. One caveat is that all of these algorithm deal with only one-dimensional time-series input data.

Using CNN to address multivariate time-series regression has became increasingly popular. Mode & Hoque (2020) compared CNN, LSTM, and Gated Recurrent Unit, showing that the results of the CNN model were more transferable when compared to the others. According to experimental analysis undertaken by Antsfeld et al. (2020), utilizing LSTM layers along with additional convolutional layers, for regression can provide a significant boost in forecasting performance. And Mehtab et al. (2020) compared three models of CNN and four models of LSTM (regression problem), and concluded that the execution time of the CNN models were 3-10 times faster than the others, while the best accuracy was achieved by one of the CNN models.

## 2 PROBLEM STATEMENT

We are given a set of multi-dimensional time-series, each with a continuous label-vector representing usage. Our goal is to predict the label of a query time-series, that is its future increase or decrease in usage. There is an imbalance in this model, in that the maximum possible decrease is bounded by $100\%$, while the maximum possible increase is unbounded. To address this, let $P_A$ be the average usage in the observed time-frame, and $P_B$ the average usage in the time-frame to be predicted. Setting $\delta = \frac{P_B}{P_A} - 1$, we define our usage increase function as:

$$f(\delta) = \begin{cases} 1 - \frac{1}{1+\delta}, & \text{if } \delta > 0 \\ \delta, & \text{otherwise} \end{cases} \tag{1}$$

The above function maps the increase or decrease onto a real value in the range $[-1, 1]$. Values tending to -1 imply significant decrease in usage, values close to 0 imply no change in usage, and values close to 1 imply extreme growth. (See also its use in Section 3.1 below.)

## 3 DATA AND PROCESSING

Before presenting the model, we describe our data sets and their select features, and detail our data processing. The data processing includes the creation of a modified label-vector for the time-series, which will be necessary for running the model.

### 3.1 DATA SETS AND FEATURES

Our main data-set is a proprietary set provided by Cloudinary, which is a software-as-a-service (SAAS) company providing media end-to-end solutions in the cloud. The data contains daily records of the media transformations undertaken by each customer, as well as delivery over the network, recorded as delivery requests and utilized bandwidth. Another important feature is the total data storage used by each customer through uploading media to Cloudinary's distributed shards; this includes original assets and assets derived via transformations. The data-set also records each customer's subscription plan – these can be standard or custom, and monthly or annual. Note that a customer in this data set is often a company consisting of many distinct users.

We also utilized the public Bike-Share Usage in London and Taipei Network data set (link), which is a spatio-temporal urban transport data set from a network of bike stations. London and Taipei are both very large cities with large bike-share systems. At each location where bikes are picked up, the individual bikes are tracked, so that each rental generates a digital footprint: which bike, from where, to where, for how long, at what date and time, and by whom. The data collects events from 2016 until the first half of 2020.

For both data sets, care was taken to determine which features should be extracted. We avoided sparse features, as well as pairs of correlated features, or features unrelated to usage. A detailed list of features extracted from the Cloudinary data set is found in the Appendix. From the Bike Sharing data set we extracted the following fields: Start date, bike ID, duration, and startStation ID. We have also filtered out customers who are outliers, meaning their usage of the product is atypically small or large.

## 3.2 DATA PROCESSING

For each data-set we computed the time-series associated with each individual user, and further spawned new user time-series at every one-week interval. While this could cause individual customers to appear as multiple time-series in the data set, we ensured that the same customer can appear (possibly multiple times) in only one of the train, validation or test groups.

Having segmented the data into individual time-series, we normalize each one: We first log-scale the time-series features, and then normalize each feature twice:

1. Self-comparison: Each customer feature $x_j$ is mapped to $z_j = \left(\frac{x_j - \mu}{\sigma}\right)$, where $\mu$ is the mean and $\sigma$ is the variance of this feature over this time-series.

2. Global comparison: Each customer feature $z_j$ is mapped to $w_j = \left(\frac{z_j - \mu}{\sigma}\right)$, where $\mu$ is the mean and $\sigma$ is the variance of this feature over this time-series.

**Labels.** It remains to calculate the usage label, as previously discussed in Section 2. We preprocess the Cloudinary data set as follows: Recall that $P_A$ is the average usage in the observation period, which we now define to be the first ten weeks of the series, that is days 1-70. For the $i$-th time-series, we create a label vector $y_i$ as follows: For entry $y_{i,j}$, we define $P_{B,j}$ to be the average usage in the 30 day period ending at day $70 + j$. We then compute $\delta_j = \frac{P_{B,j}}{P_A} - 1$ and set $y_{i,j} = f(\delta_j)$ (as defined in Section 2), and obtain the distribution of our label. See Figures 1 and 2 for an illustration of the computation of the label for $j = 60$ and the distribution of these labels in the Cloudinary data set.

The Bike-share data set is preprocessed in the same way, except that we take $P_{B,j}$ to be the average usage in the 21 day period (instead of a 30 day period) ending at day $70 + j$. Our choice for this period length reflects the fact that the behavior captured in the Bike-share data set is strongly influenced by the weekly cycle, as typically usage is determined by the week day. While the Cloudinary data set is also influenced by a weekly cycle, the standard approach in the corporate world is to compute usage in monthly blocks, making this the more appropriate measure for this data set.

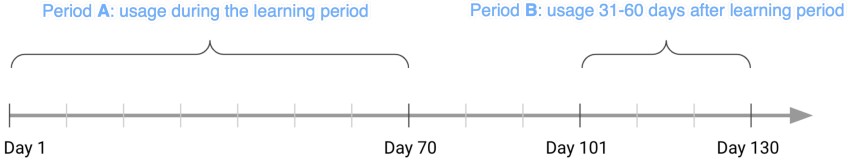

Figure 1: Label of data

## 4 CNN CONSTRUCTION

We model our data using Convolutional Neural Networks. These are known to return superior results for time-series classification when compared to other models (Canizo, 2019; Okita & Inoue, 2017; Yang et al., 2015), and when combined with auxiliary output are known to give predictions with 50% improved accuracy compared to state of the art models (Zhang et al., 2018).

Our exact model depends on the period to be predicted. In what follows, we initially describe the model for the Cloudinary data set and predicting future usage of 60 days, meaning predicting the average usage in the 30-day period including days 31-60 after the initial 70 days of the time-series. Hence we stipulated above (Sections 2,3.2) that the time-series in the training set are labelled with a scalar in $[-1, 1]$ representing the average of this period.

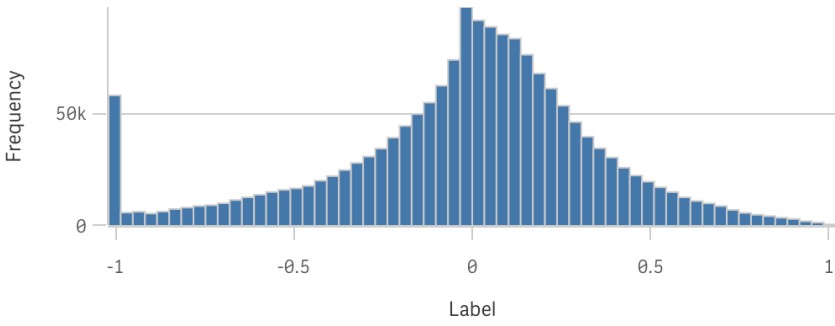

Figure 2: Label frequency

Considering a single feature in a sequence (as in Figure 3), the 1-dimensional convolution uses a kernel that weights adjacent observations, and by using pooling and a second convolution, enables the model to learn the trend of this specific feature. In our case we have multiple features, so we combine their output into a fully connected layer.

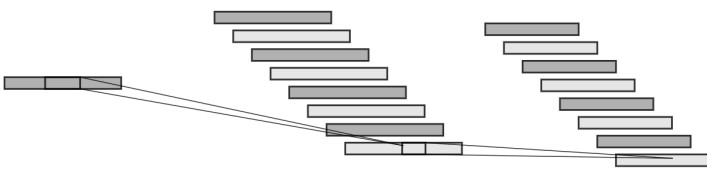

Figure 3: One dimensional convolution on a single feature

The structure of our model is as follows: It is composed of two convolutions (with 40 and 50 filters, respectively), followed by a fully connected (dense) layer of 150 neurons, using RELU as the activation. (see Figure 4). Following this unit are an additional six fully connected layers of auxiliary output (see Figure 5; the first auxiliary layer is also illustrated in Figure 4). The goal of each auxiliary layer is to predict future usage in the short term, and to pass this prediction to the next auxiliary layer. This next auxiliary layer will predict future usage for a slightly later period, until the final auxiliary layer yields a long-term prediction. In particular, we view each layer as predicting an additional ten days in advance, leading to a final prediction of sixty days. For each of the six auxiliary layers we utilize a dropout of 50% to avoid over-fitting.

We used grid search on the validation data set to determine the optimal number of convolutions, number of filters in each convolution, size of the fully connected layer, and the drop-out rate; the best results determined the parameters adopted in the above model. We also tried reducing the drop-out rate for later auxiliary level, but found that this did not yield appreciably better results. For computing the optimal number of auxiliary layers, we added layers until the improvement was negligible.

To each auxiliary layer $k \in \{1, 2, 3, 4, 5, 6\}$ we associate a mean-squared error (MSE) loss function (as in Figure 5). Recall that the motivation behind auxiliary layer $k$ was to produce a future prediction for $10k$ days ahead. Then

$$\text{loss}_k = \frac{1}{n} \sum_i \left( y_{i,10k} - \tanh(\bar{y}_{i,10k}) \right)^2 \tag{2}$$

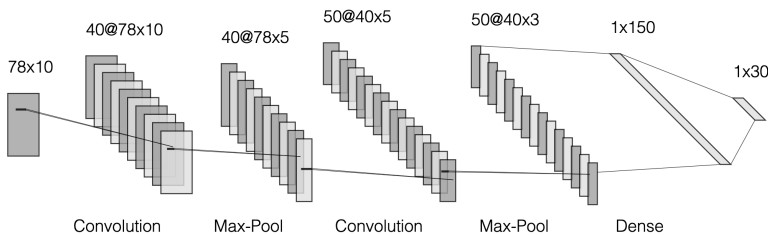

Figure 4: The first part of the model includes two layers of 1-dimensional CNN followed by a fully connected layer. At the far right of the figure is the first auxiliary layer.

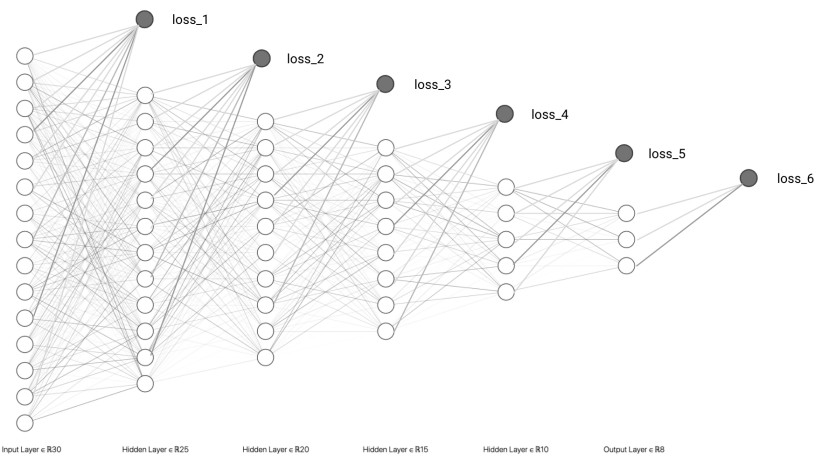

Figure 5: The second part of the model includes 6 auxiliary layers.

where $y_{i,10k}$ is the true label of future day $10k$ in time-series $i$ (as defined in Section 3.2) , and $\bar{y}_{i,10k}$ denotes the $k$-th auxiliary layer's prediction for this value. The $\tanh$ function is used to restrict the range of $\bar{y}_{i,10k}$ to $[-1, 1]$, while having only a minor effect on values already within this range.

The final model attempts to minimize a sum of the six individual loss functions. However, because each coordinate in label-vector $y_i$ is determined by averaging time-series entries over the previous 30-day period, a simple sum would overvalue the time-series entries in the earlier part of the evaluation period, and undervalue the later time-series entries in that period. For example, the time-series value of day 31 after the observation period figures into all labels $y_{i,j}$ for $j \in [31, 60]$, while the time series value of day 60 only figures only into the value of $y_{i,60}$ in this range. To address this issue, we take a weighted sum instead, and the goal becomes to minimize

$$\text{joint-loss} = \frac{1}{6} \sum_{k=1}^{6} \left( \frac{\text{loss}_k}{\delta_k} \right) \qquad (3)$$

for moderately decreasing values $\delta_k$ chosen via experimentation.

We stop the training after encountering 20 consecutive epochs without improvement in validation loss, as presented in the following algorithm:

$best\_validation\_loss\_score \leftarrow \infty$;
$remaining\_epochs \leftarrow 20$;
**while** $remaining\_epochs > 0$ **do**
    Train model one epoch and calculate $current\_validation\_loss\_score$ ;
    **if** $best\_validation\_loss\_score \geq current\_validation\_loss\_score$ **then**
        $best\_validation\_loss\_score \leftarrow current\_validation\_loss\_score$;
        $remaining\_epochs \leftarrow 20$;
    **else**
        $remaining\_epochs \leftarrow remaining\_epochs - 1$;
    **end**
**end**

**Algorithm 1:** Stopping the training process

**Extension to other prediction periods.** The above description gives our model for a 60-day prediction. In this model, each auxiliary layer corresponded to a forward prediction of 10 days. Hence, to adapt this model to different forward value predictions – specifically, 30 day and 90 day predictions – we simply modify the model to contain three or nine auxiliary layers, respectively, instead of the original six.

For the Bike-share data set, the experiment below attempts a forward prediction of nine weeks. Recall that for this data set we average periods of 21 days (instead of 30), and therefore we will view each auxiliary layer as giving a forward prediction of one week (instead of 10 days). It follows that for a forward prediction of nine weeks we can use a model of nine auxiliary layers.

## 5 EXPERIMENTS

We run the following common machine learning models on our data sets: Random forest, Fully Connected Artificial Neural Network (ANN), Recurring Neural Network (RNN), and our augmented Conventional Neural Network (CNN) model. For both data sets, we divided the data as follows: 70% of the customers in the training set, 20% of the customers in the validation set, and 10% of the customers in the test set.

### 5.1 CLOUDINARY DATA SET

Here we attempted to predict future usage of 30 days, 60 days and 90 days (meaning the respective averages of the three 30-day periods $[1, 30], [31, 60], [61, 90]$), and reported the success rates for each period in a designated table. The computed error values for each model is

$$\frac{1}{n} \sum_{i=0}^{n} \mathbb{1}_{|y_{i,j} - \overline{y}_{i,j}| > d} \tag{4}$$

where $y_{i,j}$ is the true label (of Section 3.2), $\overline{y}_{i,j}$ is the output prediction of the model, $\mathbb{1}$ is the indicator function, and $j$ takes values in $\{30, 60, 90\}$. The value $d$ is a threshold on the difference between these two models; if the difference exceeds the threshold, the prediction is taken to be an error, and otherwise the prediction is viewed as successful. This is a common regression measure, and is especially appropriate for our setting due to commercial rationale behind the model: Companies seek to maximize the customers correctly served. We tried values $d \in \{0.05, 0.1, 0.15, 0.2, 0.25, 0.3\}$ Comparisons for 30, 60, and 90 days are presented in Tables 1, 2 and 3.

The results show that our method almost always gives improved results when compared to the others. This is especially true as the forward prediction period is increased, making predition more difficult.

| Value of d | Random Forest | ANN | RNN (LSTM) | Our model (CNN) |
|---|---|---|---|---|
| 5% | 21.62% | 21.52% | 26.66% | **28.33%** |
| 10% | 39.78% | 43.22% | 47.51% | **50.40%** |
| 15% | 55.56% | 59.25% | 63.41% | **66.40%** |
| 20% | 67.20% | 70.25% | 74.80% | **76.93%** |
| 25% | 76.29% | 78.21% | 82.17% | **84.09%** |
| 30% | 82.16% | 84.37% | 87.04% | **88.90%** |

Table 1: Performance of predicting behavior at 30 days in Cloudinary data

| Value of d | Random Forest | ANN | RNN (LSTM) | Our model (CNN) |
|---|---|---|---|---|
| 5% | 16.42% | 18.01% | 18.27% | **19.09%** |
| 10% | 32.73% | 34.66% | 35.52% | **36.17%** |
| 15% | 47.10% | 48.45% | **50.96%** | 50.88% |
| 20% | 59.77% | 59.92% | 63.29% | **63.73%** |
| 25% | 69.44% | 70.77% | 73.34% | **74.57%** |
| 30% | 77.92% | 80.13% | 80.84% | **81.71%** |

Table 2: Performance of predicting behavior at 60 days in Cloudinary data

| Value of d | Random Forest | ANN | RNN (LSTM) | Our model (CNN) |
|---|---|---|---|---|
| 5% | 15.20% | 15.68% | 14.27% | **17.09%** |
| 10% | 28.37% | 29.95% | 28.27% | **31.56%** |
| 15% | 40.16% | 42.31% | 41.03% | **45.42%** |
| 20% | 50.42% | 52.87% | 51.40% | **55.90%** |
| 25% | 59.54% | 61.52% | 60.81% | **63.65%** |
| 30% | 67.16% | 68.06% | 68.40% | **70.28%** |

Table 3: Performance of predicting behavior at 90 days in Cloudinary data

| Value of d | Random Forest | ANN | RNN (LSTM) | Our model (CNN) |
|---|---|---|---|---|
| 1% | 35.14% | 45.68% | 53.87% | **59.50%** |
| 2.5% | 74.79% | 82.31% | 88.22% | **92.20%** |
| 5% | 82.14% | 87.41% | 95.81% | **98.49%** |
| 7.5% | 95.03% | 96.52% | 98.61% | **99.18%** |
| 10% | 97.87% | 98.06% | 99.47% | **99.86%** |

Table 4: Performance of predicting behavior at 9 weeks in Bike-share data

## 5.2 BIKE RENTAL DATA-SET

For this data set, we attempted to predict the total duration of usage in each station. As before the training period was 10 weeks (weeks 1-10), but now we attempt to predict usage at 9 weeks (week 19). As above, the computed error is taken to be $\frac{1}{n} \sum_{i=0}^{n} \mathbb{1}_{|y_{i,j} - \overline{y}_{i,j}| > d}$, but with $d \in \{0.01, 0.025, 0.05, 0.075, 0.1\}$. The lower values of $d$ taken here in comparison to those taken for the Cloudinary data set are due to this data set being easier to predict – indeed, the practices of individuals as recorded in the bike-share data set are simpler and more predictable than those of the clients in the Cloudinary data set, which are usually large companies of many users. This is also the reason we sufficed here with only a single experiment, utilizing the most advances (nine layer) model. The results are reported in Table 4.

While all methods performed well for high values of $d$. our method returned better results. For low values of $d$ our method is appreciably better.

## 6 CONCLUSIONS

We attempted to predict future customer behavior using multi-dimensional time-series as our input. We utilized two data-sets: The first was the Cloudinary data set, where customer are often large companies. Our second data set was a public data set containing behavior of individuals using the Bike-share in London and Taipei. We suggested a CNN-based model with auxiliary output, and compared this model to other standard approaches. Our approach achieved the best results in almost all of the comparisons undertaken. Our model performed better in the Bike-share data, and we believe this was because individual behavior is easier to predict than that of large companies.

We note that we also ran our model on less structured data, such as the Exchange rate per country data set (link). Our model returned poor results for this set, and we believe this is because its behavior is erratic, see Figure 6.

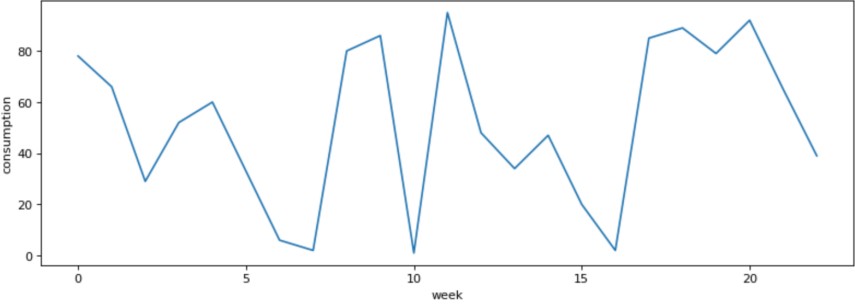

Figure 6: Exchange rate per country. Our model was unable to predict the erratic behaviour of this data set.

**Future research.** One shortcoming of our model is that it must be modified and re-trained for different forward prediction periods. Some other models (such as RNN), do not have this issue. In future work, we would like to modify the structure of our model to allow changing the forward prediction period without needing to retrain the model.

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

# 7  CLOUDINARY DATA SET FEATURES

The following features were extracted from the data:

**(1) Day of the week:** It is evident from the data that customers have a weekly consumption cycle. Some customers have usage that peaks during workdays, while others peak during weekends.

**(2) Months since customer registered:** Seasoned customers are more stable and predictable, while newer customers may still be learning and experimenting with the functionality of the system.

**(3) Utilization:** This category includes five different features, meant to give the model the ability to differentiate between different subscription plans. **(3.1) Normalized consumption:** This is the cost to the customer computed as if the customer were enrolled in the default plan. The default plan is generally suited to small customers, while large customers typically have custom (and heavily discounted) plans. **(3.2) Consumption utilization numerator:** The unit consumption of each customer based on the customer's plan, and including the consumption over the entire payment period (month or year). **(3.3) Consumption utilization plan denominator:** The number of expected consumption units based on the customer's plan. For monthly plans this is simply the number of units of the plan, and for annual plans this is the number of units used to date. **(3.4) Monthly consumption amount:** The number of units consumed by the customer in the last month. **(3.5) Consumption utilization denominator:** This is the normalized available units in the plan for the relevant month. For monthly plans this is simply the number of units included in the plan, and for annual plans this is the number of units divided by 12.

**(4) Storage:** In this category we created five different features to allow the model to adjust to the storage requirements of various file types. **(4.1) Resources:** A count of the number of media objects the customer has uploaded. **(4.2) Derived resources:** A count of the number of transformed (derived) media objects the customer holds. **(4.3) Resource storage:** The size of the uploaded resources (in Terabytes). **(4.4) Derived resource storage**: The size of transformed resources (int Terabytes). **(4.5) Backup storage TB:** Total storage backup.

**(5) Requests:** In this category we created six different features to allow the model to adapt to delivery consumption. **(5.1) Image requests:** This is a count of the number of image requests made by the customer over the Content Delivery Network (CDN). **(5.2) Video requests:** This is the same as image requests, but for video. **(5.3) Total requests:** This includes requests for all file types. **(5.4) Image bandwidth:** The size of all delivered images (in Terabytes). **(5.5) Video bandwidth:** The size of all delivered video assets (in Terabytes). **(5.6) Total bandwidth:** This includes the total sizes of image, video and other file formats (in Terabytes).

**(6) Transformations:** Under transformations we created 3 different features to help the model understand the core usage of our customer during the last month. As usage of transformation is the strongest indication of our customers experience . **(6.1) Image transformations:** Amount of transformations that were done on images in the last months, a customer that doesn't do any transformations is a strong indication that will later decrease his requests and later his storage. **(6.2) Video transformations:** Amount of video transformations done in the last month by the customer **(6.3) Video transformations:** Total amount of transformations (image, video and other)

**(7) Monetization:** Under monetization we created 3 different features that reflect the commercial affect **(7.1) Company cost:** Total company cost based on the customer usage, this includes the storage, bandwidth and transformations. Some customers the cost may be correlated to their plan, yet some may be abusing or un-abusing. **(7.2) Monthly Recurring Revenue (MRR):** The current MRR of the customer, no matter if he is in a monthly plan or annually plan. This value represents the recurring revenue the customer pays for the all of the services he gets from the company. **(7.3) Overages dollar:** In case a customer exceeded his plan he starts to pay overages, most customers are fine with paying overages especially when they are small, yet some can get a bill shock in case the amount is very high which may lead to future churn.

