# OpenReview forum: "Predicting subscriber usage: Analyzing multi-dimensional time-series using Convolutional Neural Networks"
_ICLR.cc/2022/Conference — ICLR 2022 Submitted_

### Official Review · Reviewer_Cb5i · 2021-10-28

**Correctness:** 2
**Technical Novelty And Significance:** 2
**Empirical Novelty And Significance:** 2
**Recommendation:** 3
**Confidence:** 3

**Main Review:**

The paper presents an empirically effective methodology to predict the product usage of a given consumer using multivariate time series.
There are two main problems that make the paper not ready.
First, the paper would have been more suitable for an industrial track or an applied analytics tracks rather than for a research track.
Indeed, in my opinion, the paper lacks novelty as it can be reduced to applying a CNN-based approach to the problem of time series regression.
Second, the paper lacks a formalism that prevents clarifying the data dimensionality. Indeed, there is a continuous discrepancy between the terms multivariate time series for the data and one-dimensional CNN for the model. Perhaps a formalization of the data and of the model would have clarified this ambiguity and helped in better understanding the data description and data preprocessing steps. Also, the overall presentation would benefit from a toy example.


**Summary Of The Paper:**

The paper presents a deep learning model to predict the product usage of a given consumer based on historical usage.

**Summary Of The Review:**

Probably the wrong venue.
Lack of formalism in the presentation creates confusion between data and model dimensionality.

---

### Official Review · Reviewer_ANiP · 2021-11-01

**Correctness:** 2
**Technical Novelty And Significance:** 2
**Empirical Novelty And Significance:** 1
**Recommendation:** 3
**Confidence:** 4

**Main Review:**

The paper is focused on an interesting topic of multivariate time-series prediction, which is a vital area surrounding many real-world application domains (besides the consumer behaviour forecasting in this paper, the applications can also pertain to a multitude of very different tasks such as wind power forecasting, weather forecasting, stock price predictions etc.). The proposed model in this paper can easily be adapted to other tasks besides the ones discussed in this paper in the opinion of this reviewer, which can be beneficial to other domains.  The paper is well written and easy to follow.

However, this reviewer believes that the proposed methodology in this paper has limited novelty - as multi-dimensional time-series prediction for long has been an area of active research in the AI community. Additionally, only utilising auxiliary inputs to make the conventional CNN architecture deeper for which the authors add 6 auxiliary layers) does not yield major  performance improvements in multiple cases as per the results discussed in the paper in Tables 2 and 3 (for the Cloudinary data) and Table 4 (for the bike-sharing data).  However, the reviewer notes that in case of the 30 day predictions with the Cloudinary dataset and when low values of ‘d’ were utilised with the bike sharing data, there were significant performance improvements. For a performance improvement in accuracy as described in the paper which mostly levels around 2-3% in most of the cases (and maximum around 6%) - this reviewer does not see any key improvements compared to the LSTM architecture the authors mention in the baseline. Also, there are no details of results obtained with the conventional CNN architecture (without using auxiliary information) in the paper - which places uncertainty into the suitability of using auxiliary information for the learning task discussed in this paper. As the authors are not comparing their proposed model with a baseline CNN, there is an uncertainty in their claim that the performance boost can only arise from incorporation of the auxiliary layers. It may well be possible that appropriate hyper-parameter optimisation and tuning of a baseline CNN  (which is generally simpler and less computationally expensive) could yield similar (or even better) results, but since it was not compared to in this study, the methodology pertaining to development of an improved model architecture in Section 4 (in comparison to conventional CNNs) in the paper is not well-supported in terms of the results.  Additionally, the literature review in the paper is very limited and mostly does not cover broader literature in multi-dimensional time-series prediction beyond the CNN utilised in the paper. This includes aspects concerning prediction of bike sharing demand by other related studies which have used very similar datasets and applied AI models (including RNNs and simpler models such as ANNs) - such as :

- Pan, Yan & Zheng, Ray & Zhang, Jiaxi & Yao, Xin. (2019). Predicting bike sharing demand using recurrent neural networks. Procedia Computer Science. 147. 562-566. 10.1016/j.procs.2019.01.217.
- Thirumalai et al. (2017). Bike Sharing Prediction using Deep Neural Networks. JOIV : International Journal on Informatics Visualization. 1. 83. 10.30630/joiv.1.3.30.

etc .  Similarly,  for the Cloudinary case study - there is no related work covered to provide an overview of related studies utilised in datasets from software-as-a-service (SAAS) companies and the past literature in applying AI for predicting subscriber usage with closely related datasets. This limits the literature review’s scope to be very broad and not particularly domain-specific in relation to the two case studies the paper focuses on.

A comment which the reviewer would like to make is regarding the computational cost (this is not critical to the paper’s argument though) - When making the network deeper by incorporating 6 auxiliary layers - it would have been useful to see information regarding the computational time (across the baselines as well as the proposed approach), which the paper does not discuss. If the computational time of the model is significantly higher than the baselines (particularly the conventional LSTM), it seems unclear as to whether the limited performance gain at the cost of high computational time (which likely may be the case on making the network deeper in the opinion of this reviewer), is justified in terms of the contribution of the approach. While this reviewer understands that in certain application domains, even a small improvement in accuracy could be potentially significantly important, the paper does not discuss the role of the ‘accuracy gain’ in relation to the two case studies, and without a clear summary of the importance of this performance boost to the domain-specific problem, which may have been an interesting result to see.

This paper would have benefited from a clearer description of the datasets, its potential challenges and the key drawbacks of the baselines  in specific cases of experiments when the performance gain in the proposed approach is minimal (particularly with the Cloudinary dataset). Specifically, a discussion of the potential applications of the specific model architecture proposed by the authors on beyond the two datasets and its usage outside these domains would have made this paper’s novelties much clearer.  Additionally, the paper should provide a summary of the key limitations of the approach and any concerns it raises in real-world deployment of the model - in comparison to simpler, easier to deploy and train, and generally computationally less expensive architectures like conventional CNNs.

It would have been useful to see a clearer conclusion in the paper on why the authors consider the proposed methodology to be well suited for the specific problem (and datasets) under consideration. Additionally, it would be nice if the authors can comment on other potential use cases of the methodology.


**Summary Of The Paper:**

This paper focuses on a multi-dimensional time-series prediction task through deep learning - particularly towards predicting consumer behaviour. The authors utilised two real-world datasets (Cloudinary and bike-sharing dataset, with the bike-sharing data being publicly available) for predicting subscriber usage to identify consumer behaviour. Additionally, the authors also mention that experiments were also conducted on a publicly available exchange-rate dataset for time-series forecasting. The key theme in the proposed approach is the incorporation of auxiliary inputs to a CNN model (by making the conventional CNN deeper with multiple auxiliary layers), and ultimately utilising the model for time-series prediction. The paper, with multiple experiments conducted across baselines (such as ANNs, Random Forest and LSTMs), highlights that forecasting individual consumer behaviour (based on the bike-share dataset) yields better results when utilising auxiliary layers in the CNN model.


**Summary Of The Review:**

Overall, while the paper’s topic is interesting - given that multi-dimensional time-series prediction is an active area of research interest in the AI community, as well as focuses on predicting subscriber usage based on two real-world datasets (of which one is publicly available),  the paper’s methodology has limited novelty. Also, the literature review is not thorough, and the paper is not well-supported in terms of clearer description of results beyond the models’ accuracies as described in the main review.

This reviewer believes that this paper would be better suitable for publication in a more applied venue - as the contribution to technical novelty, particularly the methodology proposed and the application of the study is very limited.  Additionally, the results should be incorporated with additional details on computation time and metrics beyond accuracy, which is integral for comparison of (deep) ML architectures with more traditional baselines, particularly as the paper aims to make the conventional CNN architecture deeper by adding additional auxiliary layers.

---

### Official Review · Reviewer_BJUT · 2021-11-02

**Correctness:** 3
**Technical Novelty And Significance:** 2
**Empirical Novelty And Significance:** 3
**Recommendation:** 3
**Confidence:** 5

**Main Review:**

Details have been given for datasets, data processing. Improvement has been shown on real datasets.

Here are a few weakness points to the paper:

1. The novelty of this paper is limited. The proposed method is essentially a variant of convolutional nets on time series data. The major difference is that it introduces multiple losses to incrementally predict multiple variables from short term to long term.

2. In the result section, it would be more convincing if the authors could provide more analysis on the improvement. Also, the authors could consider studying the effect of multiple losses, e.g., by comparing to a CNN without (or with less) auxiliary layers

3. Some figures are not informative. For example, more details should be added to Figure 3 and its caption to make it clear.

4. The authors compared the proposed method to standard ML approaches, but missed some state-of-the-art baselines. Also, it would be better to discuss/compare with other approaches used in weather forecasting, which is known to be a challenging forecasting problem with chaotic dynamics nature and long-term patterns.

5. The presentation needs to be polished. Typos: Page 1, “to accurate to accurately predict”


**Summary Of The Paper:**

This paper is focused on the development of an automated model for predicting multi-variate future usage. This is an important problem for industrial companies that rely on subscriber systems. Long-term future prediction/forecasting is known to be a challenging problem, especially for systems with complex dynamics and high dimensional variables. In this paper, the authors propose a CNN-based model with multiple outputs/losses. Multiple auxiliary layers are constructed to incrementally produce predictions from short-term future to long-term future. Results have shown some improvement on two datasets.

**Summary Of The Review:**

Overall, this paper studies an important problem, but the novelty is limited. The experiments are also insufficient.

---

### Official Review · Reviewer_Y6ay · 2021-11-02

**Correctness:** 3
**Technical Novelty And Significance:** 2
**Empirical Novelty And Significance:** 2
**Recommendation:** 3
**Confidence:** 5

**Main Review:**

While the proposed techniques aren't identical to methods in the literature (to the best of my knowledge), there is significant overlap with well-known methods that lack citation. Notably, the paper doesn't cite Temporal Convolutional Networks and its variations (other than briefly alluding to commercial implementations), which has been used extensively in similar scenarios. TCNs share many of the properties of the proposed method, including the multi-scale temporal resolution.

For reference, here is a paper applying TCNs to a comparable time series forecasting task: https://www.nature.com/articles/s41598-020-65070-5

With the lack of contextualization in the relevant literature, the paper also misses important baselines for its two experiments. The proposed method beats vanilla techniques (namely, vanilla RF, ANN, CNN, and RNN), but it's unclear if the method is competitive with state-of-the-art techniques (like TCN and others). Additionally, for such a task, it's helpful to understand how much the additional features help beyond an autoregressive effect; it'd be helpful to also have included a simple autoregressive model in the baseline.

The paper transforms the labels to a number in [-1, 1] based on rates of change, which it reasonably justifies for modeling. However, the metric (|y - y'| < threshold, after the transformation) used in the experiments is not very intuitive and could be misleading, especially considering the paper focuses on business applications. For example, the real-world difference for transformed values 0.999 and 0.9999 is massive, but the error metric will be minor. It'd be helpful to include some standard metrics (e.g. MAE and RMSE) on the original label values (before the transformation), for contextualization.

The paper is otherwise clear and generally well-motivated, and covers an important business use case. I appreciate that the authors used a public dataset in addition to their private dataset.


Other thoughts:

- The paper mentions multiple categorical features (e.g. plan type), but does not specify if there was any special treatment for them. Are they one-hot encoded, treated as integers, ...?
- [nit] Typo: Section 3 -- "and their select[ED] features,"
- [nit] Style: mixed usage of "data sets" and "data-sets" (hyphen)



**Summary Of The Paper:**

The paper presents a model architecture for time series forecasting based on CNNs. The proposed method incorporates multi-dimensional inputs as well as auxiliary outputs for different time scales. Compared to the demonstrated baselines, the paper's method generally outperformed across two datasets for product usage (cloud SaaS and bike rentals).


**Summary Of The Review:**

My primary concerns are missing literature (and therefore baselines), and unintutive evaluation metrics.

---

### Official Review · Reviewer_4yDn · 2021-11-04

**Correctness:** 3
**Technical Novelty And Significance:** 2
**Empirical Novelty And Significance:** 2
**Recommendation:** 3
**Confidence:** 5

**Main Review:**

This paper is an application paper and thus lacks methodology and theory strengths.

In the contribution part of the introduction section, it shows the proposed method highly resembles the well-known iterative method for multi-step predictions. The proposed idea of using auxiliary layers is straightforward and lacks technical motivation and insights.

In the related work section, the listed papers are less representative and many relevant and technically advanced work are missing, for example (but not limited to),

2019, NeurIPS, Think Globally, Act Locally: A Deep Neural Network Approach to High-Dimensional Time Series Forecasting
2019, ICML, Deep Factors for Forecasting
2021, ICLR, Multivariate Probabilistic Time Series Forecasting via Conditioned Normalizing Flows
2021, ICML, Autoregressive Denoising Diffusion Models for Multivariate Probabilistic Time Series Forecasting
2021, ICML, RNN with Particle Flow for Probabilistic Spatio-temporal Forecasting

The presentation of this paper needs to be re-structured for better highlighting the methodology and theoretical contribution.
The author is supposed to clarify the special characteristics of the data if any, given that the data processing is in one of the main sections in the paper, while the content is ordinary and less interesting. The section presenting the models unnecessarily includes too many implementation details, e.g. network setups, hyper-parameter search, which are rather trivial and should be put in the experiment or appendix sections.

In the experiment, the author used a less common metric involving an error threshold parameter, while the commonly used point and probabilistic errors are missing, e.g. mean squared error, quantile error, test likelihood, etc (refer to the papers listed above). The authors are expected to elaborate more on the motivation of the chosen metric. It would be better if the authors present the results of the common metrics as well, because existing methods are mostly evaluated on these metrics.

Even for the chosen metric, the result is confusing in that based on the error definition Eq.(4), for an error threshold, the interpretation would be the less the metric value, the better the performance, because it means there is less predictions different from the true values beyond the error threshold. In this sense, the proposed method is performing the worst.

The baselines are rather naïve and insufficient to justify the advantages of the proposed method.


**Summary Of The Paper:**

This is an application paper about utilizing CNN for multi-dimensional time series from subscriber usage. The authors proposed to use auxiliary layers to produce multi-step predictions. Overall, this paper lacks technical and theoretical novelty. The presentation needs significant improvements and the experiment section is unsolid to justify the proposed methods.

**Summary Of The Review:**

On the methodology side, the proposed method is a stack of ordinary convolutional layer and auxiliary output layers and thus falls short of strong motivation, novelty and insights. On the presentation side, it is more a technical report and lacks the differentiating elements from existing works. The experimental evaluation is weak, because of the less common error metrics, naïve baselines, and unconvincing result analysis.

---

### Decision · Program_Chairs · 2022-01-20

**Decision:**

Reject

**Comment:**

ICLR is selective and reviewers are not sufficiently enthusiastic about this paper. In particular, they point out closely related methods that should be cited and compared to as baselines. The reviews are of good quality, and the authors did not respond.